# Experiences and Perspectives of GC-MS Application for the Search of Low Molecular Weight Discriminants of Schizophrenia

**DOI:** 10.3390/molecules28010324

**Published:** 2022-12-31

**Authors:** Natalia Porozova, Elena Danilova, Igor Senshinov, Andreas Tsakalof, Alexander Nosyrev

**Affiliations:** 1Laboratory of Analytical Toxicology and Immunochemistry, Department of Biomedical Problems, National Research Center on Addictions, Branch of Serbsky Institute for General and Forensic Psychiatry, 119839 Moscow, Russia; 2I.M. Sechenov First Moscow State Medical University, 119991 Moscow, Russia; 3Department of Analytic, Chemistry, Faculty of Chemistry, Lomonosov Moscow State University, 119991 Moscow, Russia; 4Laboratory of Biochemistry, Faculty of Health Sciences, School of Medicine, University of Thessaly, Biopolis, 41111 Larissa, Greece

**Keywords:** psychiatric disorders, schizophrenia, GC-MS, biomarkers, schizophrenia diagnosis

## Abstract

Schizophrenia is one of the most severe chronic mental disorders that is currently diagnosed and categorized through subjective clinical assessment of complex symptoms. At present, there is a recognized need for an objective, unbiased clinical test for schizophrenia diagnosis at an early stage and categorization of the disease. This can be achieved by assaying low-molecular-weight biomarkers of the disease. Here we give an overview of previously conducted research on the discovery of biomarkers of schizophrenia and focus on the studies implemented with the use of GC-MS and the least invasiveness of biological samples acquisition. The presented data demonstrate that GC-MS is a powerful instrumental platform for investigating dysregulated biochemical pathways implicated in schizophrenia pathogenesis. With this platform, different research groups suggested a number of low molecular weight biomarkers of schizophrenia. However, we recognize an inconsistency between the biomarkers or biomarkers patterns revealed by different groups even in the same matrix. Moreover, despite the importance of the problem, the number of relevant studies is limited. The intensification of the research, as well as the harmonization of the analytical procedures to overcome the observed inconsistencies, can be indicated as future directions in the schizophrenia bio-markers quest.

## 1. Introduction

Schizophrenia is one of the most severe chronic mental disorders with still undisclosed etiological mechanisms that usually develop slowly over months or years and requires lifelong treatment. The critical age of onset of schizophrenia is 15–25 years for men and 25–30 years for women [1]. It is associated with many psychiatric (depression, addictive disorders, cognitive impairment) and somatic comorbidities (hyperprolactinemia, sexual dysfunction, cardiovascular disease, diabetes, high blood pressure) [2,3]. According to the World Health Organization, schizophrenia affects not more than 1% of the population; however, this disorder is still associated with the highest median societal cost per patient (mean USD 18 313 per year) among all mental disorders [4,5,6]. However, an early diagnosis of schizophrenia contributes to its successful treatment that slows down the progression of the disease, might potentially reduce hospitalizations and increase independent living by 41.5–46%, as well as raise employment by 42–58% [7]. At present, there is a recognized need for a highly accurate, unbiased clinical test for schizophrenia diagnosis at an early stage and categorization of the disease. Detection and application of disease-predictive biomarkers is currently the most intensively developing approach. The biomarkers are thought in different biological samples such as plasma, serum, urine, cerebrospinal fluid (CSF), hair, sweat and recently exhaled breath. Based on the pathogenesis of any disease, two groups of molecular biomarkers can be distinguished. Markers of one group are present in the body both normally and in disease, but in different quantities. Markers of the other group are unique to the disease due to specific to the particular disease metabolic aberrations. The latter are the most valuable and, if found, can clarify the biochemical underpinnings and pathogenesis of schizophrenia. Metabolite markers studies have already contributed to the formulation of several hypotheses for the pathophysiological origin and development of this disease, e.g., PUFA or energy metabolism hypothesizes [8,9]. Currently, there is less hope of finding a single specific marker for schizophrenia, but rather a specific characteristic pattern/profile of a number of metabolites’ biomarkers. The modern methods of instrumental and bioinformatic analysis support this approach, and in this systematic review, we analyze the studies for the detection of molecular biomarkers of schizophrenia conducted between 1969 and 2021 with the use of chromatography–mass spectrometry techniques, highlight the most important results acquired and outlined future perspectives. Special emphasis is given to the investigation of metabolites by gas chromatography–mass spectrometry. By GC-MS wide range of volatile and semi-volatile compounds can be analyzed as well as inherently non-volatile compounds that can be volatilized by chemical derivatization [10]. GC-MS is the method of choice for the targeted or untargeted analysis of volatile metabolites excreted by the organism, which are intensively investigated nowadays for the development of non-invasive methods of preliminary screening and early disease diagnosis [11,12]. The main advantages of capillary GC-MS are high peak capacity, high-resolution power, low detection limits and tunable selectivity of MS detector, e.g., high selectivity in selected ion monitoring mode. Moreover, electron impact ionization, usually used in GC-MS applications, generates highly reproducible mass spectra that can be directly used for identification of the compound by comparison to library spectra with the use of spectral matching algorithms, e.g., NIST/EPA/NIH mass spectral library with 308,869 compounds spectra. 

## 2. Methods Used for Literature Search and Publications Selection 6993452864

The NCBI PubMed, Google Scholar, ScienceDirect and Semantic scholar databases were searched for original experimental articles using the keywords schizophrenia or psychosis, GC-MS or gas chromatography–mass spectrometry or biological marker or biological signature. Three of the chosen machines supported the Boolean advanced search, so the keywords mentioned above were formulated accordingly to search machine rules. A semantic scholar has different mechanics for relevance search, so mentioned keywords were chosen, respectively, to logical operators. 

The following criteria were defined for the inclusion of papers in this study:
The work should be an original study (not a review) involving patients with schizophrenia (or its various forms) and a control group of individuals without mental illness. Biomarker studies of other psychiatric diseases to schizophrenia as a comparison group would also be of interest to differentiate these illnesses. The diagnosis of schizophrenia should be made based on DSM-III-R or DSM-IV (V).Biological samples should be represented by biomaterial available for minimally invasive acquisition and suitable for early diagnosis (plasma/serum, urine, saliva, sweat, heart, exhaled air). Some works using cerebrospinal fluid were of interest as demonstrating the potential of GC-MS but, strictly speaking, did not meet the criteria of this review due to the invasiveness of the intervention and were referred to as background information. Post-mortem studies, animal brain tissue, and human cadaver studies were excluded due to possible post-mortem changes.The original articles were selected so that individuals preferably represented the group of patients with schizophrenia with a first episode or with suspected schizophrenia before receiving therapy (drug-naïve schizophrenia, hereafter referred to as T0). However, the treated cases were also of interest (Tn, where n is the duration of therapy in months or relapse of schizophrenia without indication of the period of illness) as a comparison group. A control group of persons without psychiatric disorders should be present regardless of whether thT0 or Tn represents the patient group.The substances of endogenous origin/products of human metabolism should be the subject of the work; research on pharmaceuticals and narcotic substances is beyond the scope of this review.The main analytical method used in studies should be gas chromatography with mass spectrometric detection (GC-MS), irrespective of ionization types of mass spectrometers. Studies involving other methods (biochemical, immunochemical, etc.) were considered as reference or complementary approaches.The age of the articles was not limited, but the study did not include papers whose data are presented only by title and in which at least an abstract is not available.Thus, processes of the studies identification were performed (Figure 1). In total, 34 papers were considered to meet best these systematic review criteria.

## 3. Results

### 3.1. The Peculiar “Smell of Schizophrenia” and Search for Volatile Organic Biomarkers of Schizophrenia

Specific smells of the body or body excretions (urine, feces) have been used for centuries by physicians to diagnose up to 40 diseases since Hippocrates, who pointed out the diagnostic value of body odors and described several disease-specific odors in patients’ exhaled air, urine and sputum [13]. At present, it is realized that disease-specific smell is due to the presence of specific odorants, which are the products of abnormal biochemical processes caused by the disease. Additionally, in many cases, the specific biochemical mechanisms of these processes have been revealed. The peculiar “smell of schizophrenia” was noticed in psychiatric hospitals as early as the 19th century [14]. In 1960, Smith and Sines showed that the sweat of patients with schizophrenia could be distinguished from the sweat of other patients and healthy individuals by smell, using specially trained rats (*p* = 0.0001) and humans (*p* = 0.005) [14]. Later, in 1969, the substance responsible for the specific odor was isolated from the sweat of schizophrenic patients and identified as *trans*-3-methyl-2-hexenoic (TMHA) acid by gas chromatography, mass spectrometry and NMR [15]. In 1971 the biochemical pathways for the formation of this odorant from 6- or 2-hydroxydopamine were hypothesized [16]. In schizophrenia patients, a defective gene encoding dopamine-b-hydroxylase promotes excessive tyrosine biosynthesis (a dopamine precursor amino acid), resulting in high dopamine levels. Since TMHA is a metabolite of 6-hydroxydophamine, its presence in patients with schizophrenia may be explained by a mechanism of self-oxidation of excess dopamine to 6-hydroxydophamine. It has been suggested that among the consequences of this self-oxidation, there may be a degeneration of peripheral sympathetic nerve endings leading to a markedly prolonged depletion of noradrenaline. According to the authors, this hypothesis is supported by cases of human exposure to phenylethylamine derivatives, similar to 6-hydroxydophamine, which produces hallucinations.

However, subsequent studies failed to confirm TMHA as a specific biomarker of schizophrenia. In 1970 Thomas L. Perry et al. did not detect TMHA in the sweat of 11 schizophrenic patients and 10 healthy controls [17]. The participants in the study were washed and wiped off, then put on plastic bags covering their entire body up to their neck and warmed by an electric heater located nearby. Thus, 100–250 mL of sweat was collected in the bag for examination by gas chromatography equipped with dual columns and a dual flame ionization detector after TMHA isolation by liquid–liquid extraction or steam distillation. The authors attributed their failure to detect TMHA to the artifacts originating from plastic bags or reagents used and presented in the treated samples. Moreover, the low resolution and selectivity instrumentation used (packed column and FID detector) could prevent chromatographic separation of TMHA from high-intensity artifacts peak and its detection. In contrast, a little later, Gordon et al. detected TMHA in the sweat of both schizophrenic patients and healthy individuals in comparable amounts and at low ppb levels [18], but only after acid derivatization to n-butyl *trans*-3-methyl-2-hexenoaten-butyl. The authors failed to detect a free acid in the sweat of normal and schizophrenic subjects by the GC-MS methodology used. The main conclusion of the study is that there is no obvious correlation between TMHA and schizophrenia. At present, it is recognized that TMHA is the most dominant and characteristic component of human axillary odorants produced from an excreted precursor with the implication of the skin microbe *Corynebacterium striatum* [19]. Poor personal hygiene characteristics for patients with schizophrenia [20] may be the reason for their excessive smell. In the reviewed above studies [17,18], the patients and healthy controls were washed and wiped off before sweat collection. The story of TMHA identification as a schizophrenia biomarker is an interesting example of how exogenous compounds can be mistakenly taken as a biomarker of disease while it is only a sign of poor personal hygiene. 

Disease-specific odorants used for centuries for disease diagnosis are only a small fraction of volatile organic compounds excreted by the organism. The not odorous volatile compounds cannot be detected by human or animal olfactory systems, but they can be detected and quantified by instrumental methods. The GC-MS is a method of choice for the analysis of volatile organic compounds (VOCs) and, in particular, the volatile organic metabolites (VOM). The total of VOMs comprises the human volatolome that is considered to present the real-time snapshot of the organism’s physiological and biochemical status. At present, the investigation of VOM is at the forefront of the development of a non-invasive method for a plethora of disease diagnosis, treatment monitoring and preventive screening [21,22]. However, it should be noted that despite the physiologically and biochemically well-grounded hypothesis of this diagnostic approach and intensive research for more than two decades, only a few VOCs based tests have been introduced in clinical practice for disease diagnosis. The main challenges are low and variable concentrations of distinctive VOCs that can be influenced by multiple factors (e.g., comorbidities or sampling and sample preparation techniques used), technical complicity of the methodology and lack of its standardization. 

Our search revealed only a few investigations of the schizophrenics’ volatolome for the detection of the disease and discrimination of the patients from healthy people or patients with other mental health disorders.

In 1993 Philips et al. investigated exhaled VOC in 88 subjects, of whom 25 were schizophrenic patients with exacerbations of psychosis, 26 were with non-schizophrenic mental disorders other than schizophrenia, and 37 were healthy volunteers [23]. The volatile organic compounds from 10 L of alveolar breath were captured in an adsorptive trap containing activated carbon and a molecular sieve. The captured VOCs were then thermally desorbed from the trap and analyzed by gas chromatography with an ion-trap mass spectrometer. The detected compounds were identified by their mass spectra and quantified with the use of pure standards. The three groups were compared based on their alveolar gradients—compound concentration in alveolar air minus concentration in inspired (environmental) air. Mean alveolar gradients of pentane, carbon disulfide (CS2), benzene, 2-methylbutane and tetrachloroethane were significantly higher in schizophrenic patients than in healthy controls, but only CS_2_ was significantly increased in comparison with patients with other mental disordered. It has been suggested that the gut microbiota may be the source of CS_2_, a known neurotoxin that may be involved in the pathogenesis of schizophrenia. It is increasingly recognized that gut microbiota can promote the pathogenesis of schizophrenia via multiple pathways [24]. Another patient’s discriminating compound, pentane, is a product and marker of lipid peroxidation as a result of oxidative stress that is recognized as one of the pathogenetic components of schizophrenia [25]. 

Two years later, the same research group [26] using the same analytical methodology and population sample (25/26/37) detected 48 volatile organic compounds in breath samples, and 11 of them were recognized as the most informative set for the discrimination between patients with schizophrenia and other subjects. This set included 2-methylbutane, trichlorofluoromethane, 2-pentanol, pentane, dichloroethane, trichloroethane, benzene, 1-chloro-2-methylbutane, 2,3,3-trimethylpentane, 2,2-dimethylbutane and tetrachloroethane. The authors did not discuss if these compounds have an endogenous or exogenous origin. Some of these compounds are clearly exogenous (trichlorofluoromethane and other polychlorinated hydrocarbons), but some are obviously endogenously produced metabolites, e.g., 2-pentanol and pentane (in accordance with HMDB [27]). Interestingly, it has been demonstrated and justified that endogenous compounds in breath can also serve as a biomarker of the disease as “metabolic pathway-specific probes” [12,28]. It should also be noted that in contrast to their previous study, carbon disulfide was not indicated as a specific discriminating compound. The data statistical analysis was made by three different approaches. Hierarchical cluster analysis showed an imbalanced distribution of patients on the dendrogram branches, as well as presented principal component analysis scatter plot did not demonstrate separated clusters of three groups. However, based on the hand-drawn discriminant line on the graph, the authors estimate that the sensitivity of the test is 80% and the specificity 61.9%. 

In Table 1, we summarized the analyzed above studies for the search for volatile organic biomarkers of schizophrenia. 

### 3.2. GC-MS Applications for the Search of Non-Volatile Biomarkers of Schizophrenia

GC-MS is the best method for the analysis of volatile compounds; however, as mentioned above, it can be a method of choice for the analysis of non-volatile compounds after their chemical transformation to volatile derivatives. This is a well-developed procedure with a wide selection of derivatization reagents and reactions [10]. After derivatization with suitable reagent amino and carboxylic acids, carbohydrates and sugars, alcohols and phenols, catecholamines and other non-volatile compounds can be efficiently analyzed by GC-MS. 

#### 3.2.1. Analysis of Amino Acids and Their Derivatives

Amino acids are substrates and, at the same time, regulators of many metabolic pathways [29,30], including the biosynthesis and downstream effects of numerous neurotransmitters. Dysregulation of these pathways has been linked to the pathophysiology of schizophrenia, for example, by kynurenic acid, dopamine, and glutamate hypotheses of the disease. This dysregulation is reflected by the variations in the particular amino acid concentrations in the organism’s biofluids that can be measured and used for disease diagnosis. 

In 1991 Baruah et al. measured the plasma concentration of serine and glycine in the plasma of 15 drug-naïve schizophrenic (T0) patients and 15 healthy controls [31]. After amino acids, isolation by ion-exchange solid phase extraction and derivatization to N,O-bis(heptafluorobutyryl) isobutyl esters were quantified by GC-MS. It was found that serine and glycine plasma concentrations were significantly higher in T0 schizophrenia patients compared to controls. The identification of the amino acids in clinical samples was made by the compounds’ retention time and mass spectrum. The authors suggested a biochemical interpretation of their findings as resulting from the reduced activity of serine hydroxymethyltransferase in the plasma of psychotic subjects. Serine hydroxymethyltransferase is involved in the degradation of both amino acids, and its reduced activity in plasma and autopsied brains (post-mortem study) of schizophrenic patients was previously demonstrated. 

However, 20 years later, Xuan et al. found glycine concentrations in schizophrenic patients’ serum significantly lower than that in healthy controls (*p* = 0.0232) and did not identify serine as a discriminatory biomarker for these two groups [32]. In this study, metabolites were isolated by liquid-liquid extraction with a methanol–chlorophorm 3:1 mixture and silylated with BSTFA. By performing GC-MS-based metabolomic profiling, the authors revealed 22 metabolites with significantly different levels in the serum of unmedicated patients and controls. These metabolites include three amino acids, namely glycine, tryptophan and aspartate, with medium (glycine, tryptophan) to high (aspartate) individual discriminatory power. All three amino acids had significantly reduced levels in schizophrenic patients. However, tryptophan was one of the best markers for the discrimination patients before and after risperidone treatment, and for this discrimination, a combination of three top biomarkers, including myoinositol, uric acid and tryptophan, showed the best diagnostic power with AUC = 0.949.

A significantly different set of also 22 differential metabolites in the serum of unmedicated schizophrenics and healthy controls was found by Yang et al. [33], who used a sample preparation procedure for the subsequent GC-TOFMS analysis similar to Xuan et al. Most of the found serum metabolites were verified by the use of reference compounds, and only two (5-oxoproline and eicosenoic acid) were using available mass spectrum library databases. The identification and verification of the detected differentiating compounds is an important step in metabolomic studies [11], often not clearly described in the publications or not convincingly enough conducted, e.g., identification only by comparison of experimental and theoretical library mass spectra is often insufficient and can be misleading. The panel of distinctive amino acids and their metabolites comprised glutamate, aspartate, 5-oxoproline, serine, phenylalanine, cystine, 2-aminobutyrate and 2-hydroxybutyrate with levels of serine slightly increased in schizophrenic patients. However, five other products of fatty acids metabolism (glycerate, eicosenoic acid, b-hydroxybutyrate), carbohydrate metabo (pyruvate) and cystine were identified as the most distinctive biomarkers of schizophrenia, and a set of these biomarkers ensured effective classification of patients and controls with AUC = 0.945 in the training set (62 patients and 62 controls) and 0.895 in the test samples (50 patients and 48 controls). Although this study was focused on the investigation of serum samples, urine samples from the same population group were also analyzed by GC-TOFMS and ^1^H-NMR to identify additional metabolite markers for schizophrenia. The differentiating metabolites detected in urine differ from those found in serum, and this underlines the role of matrix selection in biomarkers discovery. 

The 2-piperidinec carboxylic acid was referred to by Al Awam et al. as a new distinctive serum biomarker of schizophrenia [34]. The 2-piperidinec carboxylic acid is non-encoded genetic amino acid previously indicated as a diagnostic marker of epilepsy. Al Awam et al. investigated the serum samples from 26 medicated patients and 26 matched by age and sex healthy controls with the use of Trace GC-DSQ II single quadrupole MS. Two metabolites were detected as distinctive markers for the medicated patient, namely 2-piperidinec carboxylic acid and oxo-proline. These metabolites were identified only with the use of the NIST mass spectral library by comparison of the experimental and theoretical data [34]. In contrast to previous studies [32,33], no other amino acids or their metabolites were detected as specific disease biomarkers, which may be a consequence of patients taking medications.

Plasma and serum were and still are the main matrices used in biomarker research, but hair can be considered as a new matrix for metabolomics research and disease biomarker detection [35]. This matrix has long been used in forensics for the detection of drug abuse and has definite advantages over other matrices concerning easy, non-invasive sampling, storage stability and the capability of offering retrospective health-relative information for the previous 3–4 months. Goa et al. demonstrated the suitability of this matrix for schizophrenia biomarkers research by quantification of tyramine in the hair of 95 schizophrenic patients, 90 healthy controls and 56 drug abusers [36]. Tyramine was quantified as an N-heptafluorobutyryl derivative by gas chromatography–quadrupole mass spectrometry. It was demonstrated that tyramine levels were significantly higher in the schizophrenic group compared to the controls. However, the two groups were not completely age-matched, and this may have affected the results, and there are no data on whether the schizophrenia episodes were primary, presented as relapse or remission, or whether treatment was used. It is also interesting that the levels of the metabolite were found to be elevated in drug addicts. Tyramine is a naturally occurring microamine derived from the amino acid tyrosine. Numerous data suggest that it plays an important role in diseases of the nervous system and is associated with major disorders of effect and cognition [36].

In Table 2 we summarise the reported and discussed above data on the differences found between the content of amino acids and their metabolites in biosamples from schizophrenic patients and normal controls.

#### 3.2.2. Glucose Metabolites and Energy Metabolism

Glucose homeostasis is fundamental for the maintenance of normal brain function. Glucose is mainly metabolized through the glycolytic pathway and the tricarboxylic acid cycle (TCA) to provide energy and biosynthetic intermediates and through the pentose phosphate pathway, an important branch of glucose metabolism that is involved in NADPH formation for biosynthetic processes and cellular redox balance.

In the work of Liu M.L. et al. [38], GC-MS based targeted metabolomic method was used to quantify the levels of 13 glucose metabolites in peripheral blood mononuclear cells (PBMCs) derived from healthy controls, schizophrenia and major depression subjects (*n* = 55 for each group). The authors divide samples and present results for training (35/group) and test (20/group) sets. However, specifying the test group, the authors did not inform about the specificity and sensitivity of the method. In the training set, seven metabolites were significantly increased in schizophrenia subjects relative to healthy controls, including glucose, glucose 6-phosphate, fructose, fructose 6-phosphate, glycerate 3-phosphate, succinic acid and ribose 5-phosphate. Moreover, four metabolites were significantly decreased in schizophrenia subjects relative to healthy controls, including glyceraldehyde-3-phosphate, dihydroxyacetone phosphate, glycerol 3-phosphate and citric acid. The other detected metabolites, pyruvate and lactic acid, did not differ between these two groups. On the contrary, Xuan J. et al. found that schizophrenic patients showed significantly higher serum levels of lactate compared to matched controls, whereas the two studies agree on the glucose increase [32]. Perhaps the different biological samples used, plasma by Xuan J. et al. and peripheral blood mononuclear cells by Liu M. et al., can explain these discrepancies in the results of the two studies. Returning to the work of Liu M.L. et al., it should be noted that in the test set, the differences in lactate were significant, and only glycerate 3-phosphate and ribose 5-phosphate did not differ in the drug-naïve schizophrenia and healthy volunteers’ groups. Importantly, 84.6% of glucose metabolites differed significantly in subjects with schizophrenia, while only two (15.4%) glucose metabolites (dihydroxyacetone phosphate and citric acid) differed from controls in patients with major depression in the test set. Thus, six metabolites were significantly increased (glucose, glucose 6-phosphate, fructose, fructose 6-phosphate, succinic acid, ribose 5-phosphate), and two metabolites were significantly decreased (glyceraldehyde 3-phosphate and glycerol 3-phosphate) in both the exercise and test sets (Table 3). Moreover, ribose 5-phosphate in PBMCs showed high diagnostic performance for first-episode drug-naïve schizophrenia subjects [38], while antipsychotics had only a subtle effect on the glucose metabolism pathway.

#### 3.2.3. Lipid Metabolism

Impaired lipid metabolism has long been considered a cause of schizophrenia [39]. Glen et al. were among the first to study abnormalities of fatty acid levels in red blood cell membranes and plasma by gas–liquid chromatography [40]. By comparison of the acquired data for 68 schizophrenics and 259 controls, they found that erythrocyte membrane arachidonic acid and docosahexaenoic acid were significantly lower in those with schizophrenia. They also revealed the bimodal distribution of polyunsaturated fatty acids (PUFA) levels in schizophrenic patients not present in the control group. These results were later confirmed by Sethom et al., who also found decreased levels of total PUFAs and arachidonic and docosahexaenoic acids in the erythrocyte membrane of schizophrenic patients compared to healthy controls [41]. However, they make no mention of a bimodal distribution of PUFA levels in schizophrenic patients. Indicating the contradictory nature of the previous data concerning the bimodal distribution of PUFA in schizophrenia, Bentsen et al. primarily examined this distribution in 97 acutely ill patients with schizophrenia and schizoaffective disorder and in 20 healthy control subjects [42]. The most comprehensive profile of twenty-eight species of fatty acids was determined in RBC by GC-MS-based method. Significant bimodal distribution of polyunsaturated fatty acids levels was found among patients. One-third of patients constituted a group (low PUFA, mean concentration 101 μg/g RBC) who had PUFA levels at one-fifth (*p* < 0.001) of those in high PUFA patients and healthy control. Bimodality was mainly accounted for by docosahexaenoic acid and arachidonic acid. The authors conclude that these findings may define two distinct endophenotypes of disease [42]. In a subsequent study, Bentsen et al. clinically and biochemically validated two endophenotypes of schizophrenia defined by levels of polyunsaturated fatty acids in red blood cells [43]. Clinically, patients with low PUFA levels had more severe negative symptoms of schizophrenia than those with high PUFA.

The work of Dag K. Solberg et al. [44] was a logical continuation of the longitudinal study by Bentsen et al. [43]. PUFAs and other components were examined in erythrocytes of the same patients (*n* = 55), only after 5 years of treatment, during remission. The control group consisted of 51 healthy people. The results were reproduced almost exactly, with 29% of patients having reduced levels of long-chain PUFAs and 71% having increased levels. A limitation of this work is that there is no direct indication that patients with schizophrenia were initially untreated. It is likely that all patients have been in therapy for some time.

In the aforementioned study by Xuan et al., the serum fatty acids composition was investigated, and they found lower levels of saturated (palmitic acid, C16:0; stearic acid, C18:0), monounsaturated (oleic acid, C18:1n-9) and polyunsaturated (linoleic acid, C18:2n-6) fatty acids in schizophrenic patients [32]. Palmitic, stearic and linoleic acid demonstrated high discriminating power as individual biomarkers for distinguishing between schizophrenic patients and healthy controls. Linoleic acid is a precursor of arachidonic acid, one of the most abundant fatty acids in the brain and an important factor for the function of all cells, particularly in the nervous system, immune system and vascular endothelium. Reductions in plasma linoleic acid can disturb arachidonic acid synthesis and metabolism. The decreased oleic acid levels, but not palmitic acid, stearic acid and linoleic acids, were found by Al Awam et al. in the serum of 26 medicated schizophrenic patients [34]. Except for the oleic acid, this group found other distinctive fatty acids but with rather low discriminating power, namely pentadecanoic acid (AUC = 0.72), heptadecanoic acid (AUC = 0.73) and eicosanoic acid (AUC = 0.82). 

It is interesting that impaired lipid metabolism can also be detected by examination of skin lipids by tape stripping of the skin’s very top layer, called *stratum corneum*, and the analysis of *stratum corneum* components by chromatography–mass spectrometry [45]. Smesny et al. characterized by high-performance thin layer chromatography and GC-MS the lipid profiles of the epidermis of 22 patients with schizophrenia (first episode) and 22 age- and sex-matched healthy controls [46]. The ceramides classification was differentiated by their sphingoid base (sphingosine [S], phytosphingosine [P], and 6-hydroxysphingosine [H]) or dehydrosphingosine (DS)) and connected a long-chain fatty acid (nonhydroxy (N), α-hydroxy (A), or ω-hydroxy (EO)). The main finding of the work was a significant difference in skin ceramide composition (AH, NH/AS, NP and EOS) between schizophrenic patients and controls. The ceramide fraction in skin lipid composition as a whole was significantly reduced in first-episode schizophrenic patients, while some individual ceramides (ceramides AH and NH/AS) were elevated. Ceramide metabolism is closely related to fatty acid metabolism, as ceramides usually consist of a sphingoid base and a fatty acid. In turn, changes in fatty acid metabolism (especially PUFAs) are well replicated in schizophrenia. However, the levels of free fatty acids measured in the work did not reach significant differences between groups of patients with a primary episode of schizophrenia and healthy volunteers. The authors explain these results by the fact that lipid extraction was performed from the upper stratum corneum of the epidermis, in which deficiency of some classes of ceramides causes a compensatory increase in levels of other classes, so the amount of PUFAs, in general, does not change significantly [46]. Thus, testing PUFA set in the epidermis using GC-MS as a separate method cannot yet claim diagnostic value in schizophrenia.

A GC-TOF/MS study by Yang J. et al. [33] found that serum levels of five medium and long-chain fatty acids (tetradecanoic acid, hexadecanoic acid, octadecanoic acid and eicosenic acid) were elevated in schizophrenic patients (T0), levels of eight long and medium chain fatty acids were increased in the urine of these patients, and two amino acids showed reduced concentrations in the urine of patients (Table 2). b-Hydroxybutyrate, a by-product of fatty acid metabolism, showed increased concentrations in the urine and serum of T0 patients. Levels of oleic acid, eicosenoic acid and linoleate in T0 patients (in serum) were elevated in this work compared to the control group, whereas in the previous work [32], in contrast, they were decreased (Table 2).

In a recent study, Rog et al. [47] did not find significant differences in PUFA serum levels between schizophrenic patients (40) and matched for age and BMI healthy controls. However, it is important that 95% of patients were treated with antipsychotic drugs, and this may have had an impact on the study findings. The estimated dietary intake of fatty acids was similar in the two groups. The authors also investigated the serum concentrations of fatty acids responsive GPR120 receptor that mediated the anti-inflammatory effects of omega-3 fatty acids and were implicated in fat metabolism. There was found no correlation between GPR120 and fatty acids serum concentrations in schizophrenic patients, but some correlations in healthy controls. Moreover, it is not clear if GPR120 serum concentrations reflect the receptor membrane activity. 

From the above-reviewed studies, it can be concluded that in the investigation of lipid metabolism main attention was given to the status of PUFA. However, short-chain fatty acids (SCFA) play an important role in microbiota–gut–brain axis and brain function. They are produced by gut microbiota and are involved in lipid and glucose metabolism. Acetate, propionate and butyrate are the three most common SCFAs and together constitute 85–89% of the SCFAs in human serum. Li et al. [48] compared the serum levels of butyric acid in 56 schizophrenic patients at baseline and after 24 weeks of risperidone treatment and correlated those with observed clinical symptoms. Serum levels of butyric acid were measured by GC-MS using Polyethylene Glycol-Based GC Column that gives the opportunity to analyze underivatized fatty acids. At baseline, there was no significant difference in serum levels of butyric acid between patients and 35 healthy controls [48]. However, after 6 months of risperidone treatment, patients with schizophrenia showed a significant increase in serum butyric acid levels (*p* = 0.030). These authors found a positive association between this increase and the reduction in schizophrenia clinical symptoms (r = 0.38, *p* = 0.019). Later, the authors determined the levels of all main SCFA (acetic, propionic, and butyric acids) and found significantly higher serum levels of total SCFAs, acetic in patients than healthy controls (*p* < 0.05) [49].

In Table 4 we summarize the discussed literature data on lipids ‘metabolism in schizophrenic patients and normal controls.

#### 3.2.4. Endogenous Cannabinoids (Endocannabinoids)

The endogenous cannabinoid receptor activator system, the pharmacological target of cannabis-derived drugs, also appears to be dysfunctional in schizophrenia. In order to test this hypothesis, Leweke F.M. et al. [50] investigated endogenous cannabinoids in the CSF of patients (*n* = 10) with schizophrenia and individuals without schizophrenia (control group, *n* = 11). Endogenous cannabinoids were isolated from CSF by HPLC and quantified by isotope dilution GC-MS (IDMS). Cerebrospinal concentrations of two endogenous cannabinoids (anandamide and palmitylethanolamide) were significantly higher in schizophrenic patients than in controls (*p* < 0.05). In contrast, levels of 2-arachidonylglycerol, the other endogenous cannabinoid lipid, were below detection in both groups. The findings were not related to gender, age or medication and appeared to reflect an imbalance in endogenous cannabinoid signaling contributing to the pathogenesis of schizophrenia. This work does not meet the criteria for an early diagnosis due to the invasiveness of biological sample obtaining but demonstrates one of the possibilities of the GC-MS method and also had a wide interest in the scientific community involved in schizophrenia research.

#### 3.2.5. Neurotransmitters and Their Metabolites

Monoaminergic systems, which use monoamines to provide neurotransmission, are involved in the regulation of mental processes, emotions, cognitive functions, memory, attention, agitation, etc. Moreover, monoamine neurotransmitters play an important role in the production and secretion of neurotrophins, in particular, neurotrophin-3 by astrocytes, which is important for ensuring the integrity of neurons, their normal differentiation and development, trophic support and their resistance to apoptosis [51]. The classical monoamine neurotransmitters are characterized by the presence of an amino group connected to an aromatic ring by the chain of two carbon atoms (-CH_2_-CH_2_-). All of them are derivatives of aromatic amino acids such as phenylalanine, tyrosine, and tryptophan and comprise catecholamines and indolamines. The levels of catecholamines (norepinephrine, dopamine) and 5-hydroxytryptamine and their metabolites were analyzed by HPLC with an electrochemical detector (HPLC-EC) and GC-MS in the CSF from age-matched elderly in-patients affected with senile dementia of Alzheimer type (SDAT) and chronic schizophrenia [52]. Significantly lower levels of two metabolites, 3-methoxyphenylacetic acid (*p* < 0.001) and 5-hydroxyindole-3-acetic acid (*p* < 0.02), were found in the SDAT group than in the schizophrenia group, while no significant differences were detected for other substances measured. These metabolites were measured by GC-MS and HPLC-EC concurrently, and significantly different levels (up to 15% for 3-methoxyphenylacetic acid) were found by two methods, despite the linear correlation between the results of two measurements with Spearman’s r > 0.9.

Trace amines are a group of endogenous compounds that are present at very low concentrations (usually 0.1–100 ng/g) in the central nervous system. They are structurally similar to the classical monoamine neurotransmitters and include primarily 2-phenylethylamine, m- and p-tyramine, tryptamine, m- and p-octopamine, and synephrine. The trace amines act as neuromodulators for monoamine neurotransmitters and are implicated in several psychiatric and neurological disorders, including schizophrenia [53]. The relationship between subtypes of schizophrenia, paranoid and nonparanoid, and 24 h urinary phenylethylamine (PEA) and phenylacetic acid (PAA) excretion has been studied by GC-MS quantification of trace amine PEA and its main metabolite PAA [54]. The PEA was quantified without derivatization on Thermon 3000 (polyester) stationary phase, while PAA was derivatized by pentafluorobenzyl bromide and quantified on OV-1 (polydimethylsiloxane). The study included 24 normal controls, 25 nonparanoid and 23 paranoid schizophrenic patients. Increased urinary PEA excretion was found in paranoid schizophrenics, but urinary PAA excretion did not show any significant difference between schizophrenics and normal subjects. The authors claim that these findings offer some indication that PEA may play a role in the pathogenesis of schizophrenia. 

The levels of PEA were also analyzed by negative chemical ionization GC-MS in the plasma of patients with schizophrenia (*n* = 41) and healthy control subjects (*n* = 34) [55]. The study revealed a significant increase in phenylethylamine (PEA) in schizophrenic patients, although no differences between paranoid and nonparanoid forms of schizophrenia were found. Moreover, the level of content did not change depending on the phenylalanine eaten within 24 h (an exogenous precursor of PEA) or the severity of the illness. However, the authors did not divide the schizophrenic patients’ group into those on and off therapy. It is possible that exposure to neuroleptics may have confounded the results. Thus, the results of O’Reilly et al. for plasma PEA study do not contradict earlier results obtained for the urine of patients in [54]; additionally, it further provides evidence that excess PEA may play a role in the etiology of schizophrenia [55] and could potentially be used in a neurotransmitter panel for the detection of schizophrenia. 

Bufotenin (N,N-dimethyl-5-hydroxytryptamine, 5-HO-DMT) is a serotonin metabolite and a potent serotonin agonist. Bufotenin was found to be present in urine samples of schizophrenic patients and autistic children but not in normal controls [56]. Elevated levels of bufotenine in the urine of schizophrenic patients were later confirmed by Kärkkäinen, J. (1988) (GC-MS) [57] and Emanuele E. (2010) (HPLC-MS) [58], but both studies demonstrated reduced (but detectable) amounts of bufotenine in the urine of healthy individuals. The connection between the appearance of bufotenin in the urine and psychiatric disorders is still unknown. It is suggested that this serotonin-like substance of the tryptamine class is endogenous and increased levels may be one of the links in the pathogenesis of mental illness. This substance probably cannot be an early differential marker for schizophrenia on its own, requires more in-depth research into its concentrations in different psychiatric disorders and has promise for the diagnosis of schizophrenia by molecular profiling. 

The functional analog of bufotenin is N,N-dimethyltryptamine, which also elevated concentrations in the group of patients with schizophrenia compared to controls, but there were no significant differences between patients with and without acute psychosis and between males and females [59].

Liu M.L. et al., 2014 observed a significant decrease in aspartic acid and homoserine in patients with schizophrenia, with a concomitant increase in dopamine, which is involved in neurotransmitter metabolism [37].

Summary of the discussed research data on neurotransmitters and their metabolites presented in Table 5.

#### 3.2.6. Markers of Oxidative Stress

Inflammation and associated oxidative stress are considered pathophysiological factors in schizophrenia and related disorders [60]. Markers of inflammation and oxidative stress are investigated as possible early traits of schizophrenia. However, the specificity of these biomarkers is questioned because inflammation is a non-specific response to a variety of harmful health factors—infection, harmful chemicals, and tissue damage. Moreover, there are different aspects of oxidative stress that can be evaluated by different biomarkers. Thus 2-deoyxguanosine (8-OH-2-dG) is used for the estimation of oxidative damage to DNA, malondialdehyde (MDA) for chemical modification of proteins and F2-isoprostanes (e.g., 8-iso-prostaglandin F2α/8-iso-PGF2α) are used as the best biomarker of arachidonic acid derivatization and lipid peroxidation [61]. Several research groups have linked the latter biomarker of urinary changes to schizophrenia [62,63].

Urinary 8-iso-prostaglandin F2α (8-iso-PGF2α) were analyzed by GC-MS in acutely ill drug-naïve first-episode patients with schizophrenia (*n* = 22), major depression (*n* = 18), and controls (*n* = 43) [64]. The 8-iso-PGF2α was purified by immunoaffinity chromatography, and after derivatization to the pentafluorobenzyl and trimethylsilyl ethers, the concentration was measured by GC–MS. It was found that at baseline, 8-iso-PGF2α levels were higher in patients with schizophrenia (*p* = 0.004) and major depression (*p* = 0.037). Interestingly, 8-iso-PGF2a levels in urine were further increased in schizophrenic patients after 6 weeks of treatment, suggesting an accumulation of membrane lipid damage in psychosis. It is unlikely that this increase is related to the effect of treatment, as no correlation was found between antipsychotic dosage and changes in 8-iso-PGF2a levels.

In GC-MS based metabolomics study [37], it was found that the levels of four other oxidative stress-related metabolites (hydroxylamine, pyroglutamic acid, tocopherol-γ and tocopherol-α) in peripheral blood mononuclear cells were significantly different in 69 schizophrenia subjects compared to 85 healthy controls. From 18 differential metabolites identified and related to energy metabolism, neurotransmitter metabolism, oxidative stress and others, pyroglutamic acid and tocopherol-α were identified as having the most effective discriminating value.

Another pattern of 22 deferential metabolites was found in serum [32] with differentiated levels of two antioxidants, uric acid and γ-tocopherol, which were significantly lower in schizophrenic patients than in controls. However, γ-tocopherol, the only compound found concurrently in both studies, was elevated in peripheral blood mononuclear cells [37] and decreased in the serum [32] of patients with schizophrenia.

The discussed reports on the markers on oxidative stress are summarized in Table 6. 

#### 3.2.7. Steroid Hormones and Their Metabolites

Increasing evidence from animal, preclinical and clinical studies demonstrate that steroids and particularly neuroactive steroids are important modulators of brain functions and may play an essential role in the pathophysiology and symptomatology of schizophrenia (SZ) and other mental disorders [65]. In this context, it has yet to be determined whether neurosteroids may serve as biomarkers in the diagnosis of these disorders. GC-MS remains a method of choice and a pre-eminent instrumental platform in clinical steroids investigation and especially steroids profiling in different biological matrices [66]. Capillary gas chromatography has a significantly higher peak capacity than liquid chromatography, allowing much more steroids to be analyzed in a single run and steroid profiling to be carried out.

One of the first GC-MS based study devoted to the investigation of steroid levels in schizophrenic patients demonstrated that dehydroepiandrosterone (DHEA) plasma mean concentration in 23 medicated patients (90.9 ± 61.4 nmol/L) is significantly higher (*p* < 0.001) than that in 23 age and gender-matched healthy controls (24.0 ± 17.9 nmol/L) [67]. Derivatized with heptafluorobutyric acid anhydride steroids were analyzed using a Finnigan Trace GC-MS and assayed in the negative ion chemical ionization mode (NCI). According to the authors, the acquired data demonstrate that DHEA levels may serve as a potential biological marker of psychosis. The authors also underline that their results may indicate the unknown role of DHEA in the pathophysiology of schizophrenia that should be further investigated. 

A similar GC-MS method was used for the quantification of 3α,5α-tetrahydroprogesterone (allopregnanolone) in plasma samples of eight outpatients with diagnosed schizophrenia in order to investigate a possible relation with aggressive and hostile behavior [68]. The results demonstrate a positive correlation between 3α,5α-THP plasma levels and increased aggressiveness and state hostility with Pearson coefficient r = 0.72, *p* = 0.043 and r = 0.72, *p* = 0.041, respectively. The authors underline that this is a preliminary result that suggests that 3α,5α-THP may affect aggression in humans [68].

There are multiple evidences that GABAergic signaling could play an important role in the pathophysiology of schizophrenia, and levels of 18 GABAergic steroids were quantified in the plasma of 13 adult men with diagnosed schizophrenia before (T0) and after 6-months therapy (T6) by atypical antipsychotics and compared with that in 19 aged-matched healthy controls [69]. The 18 serum steroids and polar steroid conjugates were assessed by quantified by GC-MS after steroid isolation and derivatization of oxo- and hydroxy-groups. The study demonstrates altered circulating GABAergic steroid levels in men with schizophrenia. GABAergic androsterone (3α5α) and etiocholanolone (3α5β) are reduced in men with T0 schizophrenia, but therapy with atypical antipsychotics restores their levels [69]. In addition to unconjugated androsterone, which is the most common GABAergic steroid in men, most other GABAergic steroids also tended to decrease in patients. On the contrary, conjugated isomers of 5β-pregnenolone were increased in patients [69]. This is the first study that demonstrates that serum levels of GABAergic steroids can discriminate schizophrenic patients from a healthy population and inform about the efficiency of applied therapy. 

Subsequently, the same research group, Bicikova et al., extended the number of quantified serum steroids up to 39 aiming to cover the whole metabolome of neuroactive steroids [70]. In contrast to [67], the authors did not find significantly higher DHEA concentrations in either naïve patients or after 6-month medication. Higher levels of pregnenolone sulfate and sulfated 5α- and 5β-saturated C21-steroid metabolites in the progesterone metabolic pathway and, at the same time, significantly reduced levels of 5β-reduced C19-steroid metabolites in both sexes patients were found. However, the treatment with atypical antipsychotics did not significantly alter the steroid levels in spite of the fact that it reduced, as demonstrated in the previous publication [69], the severity of schizophrenia symptoms registered as clinical global impressions scale (CGI score). This is an interesting finding that sets up the question about the mechanism of steroids’ implication in the pathophysiology of schizophrenia.

#### 3.2.8. Identification of Schizophrenia by Metabolites Set/Fingerprint/Pattern

As demonstrated above, schizophrenia is a complex disorder involving the dysregulation of different pathways, and one single cause of schizophrenia has not been identified. Dopaminergic, glutamatergic and GABAergic neurotransmitter systems can be affected in schizophrenia and lead to different clinical manifestations and different types of the disease. Thus, it is suggested that holistic metabolome profiling may reveal a set of metabolites, also called metabolites’ signatures, which may have better clinical validity and utility in diagnosing disease than biomarkers of a single metabolic pathway. Really, as demonstrated by Xuan et al., GC-quadrupole-MS-based metabolome profiling in the serum of 18 schizophrenic patients before and after 8-week risperidone therapy revealed 22 discriminating metabolites of different metabolic pathways [32]. Biomarker identification was performed with the use of commercial compound mass spectra libraries NIST, NBS and Wiley. The set of four metabolites, namely palmitic acid (lipids metabolism pathway), citrate (TCA circle), myo-inositol (inositol phosphate metabolism) and allantoin (uric acid metabolism), ensured excellent classification performance between schizophrenic patients and controls with AUC = 0.958. Excellent discrimination was also achieved between patients before and after risperidone treatment with AUC = 0.949 and with the use of three metabolites signature, namely myo-inositol (inositol phosphate metabolism), uric acid (purine metabolism pathway) and tryptophan (amino acids metabolism pathway).

Later, Yang et al. profiled serum metabolites using GC-TOF-MS and also identified 22 differential metabolites between unmedicated schizophrenia patients and normal control groups [33]. Most of the metabolites were identified with the use of reference compounds. However, these metabolites were substantially different from those earlier revealed by Xuan et al. ([32], Table 7). This inconsistency demonstrates the need for the standardization of metabolite profiling procedures because it can be the result of discrepancies in analytical procedure or data treatment, e.g., metabolites identification and statistical analysis. From 22 discriminating metabolites, Yang et al., using the Akaike information criterion (AIC), extracted the five most efficient for classification accuracy metabolites, namely glycerate, eicosenoic acid, β-hydroxybutyrate (all fatty acids metabolism), pyruvate (carbohydrates metabolism) and cystine (amino-acid metabolism). With this set of metabolites, the classification between patients and healthy controls was excellent in the training set (AUC = 0.945) and good in the test set (AUC = 0.895) [33].

Obviously, the matrix used for metabolite profiling can also affect the results of discriminating metabolites identification. Liu et al. used peripheral blood mononuclear cells (PBMCs) from schizophrenia subjects (*n* = 69) and healthy controls (*n* = 85) to identify and validate biomarkers for schizophrenia [37]. In total, 18 distinctive metabolites from different metabolic pathways (energy metabolism, oxidative stress and neurotransmitter metabolism) were revealed (Table 7). The authors did not describe in detail the way of identifying these metabolites compounds. By using the Akaike information criterion (AIC), a set of three metabolic biomarkers was identified as providing the highest degree of discrimination. This set includes pyroglutamic acid, sorbitol and tocopherol-α and ensures good classification efficiency for the training set with AUC = 0.82; however, it is poor for the test set with AUC = 0.71. 

Combined metabolome and proteome profiling of serum was undertaken by Al Awam et al. by using GC-Q-MS and MALDI-TOF-MS instrumental platforms, respectively [34]. The study revealed nine distinctive metabolites (Table 7) and estimated classification efficiency (AUC), sensitivity, selectivity and accuracy for each metabolite separately. 2-piperidinec carboxylic acid, 6-deoxy-mannofuranose, galactoseoxime proved to have the best discriminating ability between schizophrenia patients and healthy controls. Several distinctive peptides were also identified, with peptide m/z 3177 being identified as having the best discriminatory value between groups (AUC = 0.96).

The metabolic phenotype of schizophrenic patients was investigated by Chen et al. in order to discover biomarkers or a set of biomarkers for discriminating schizophrenia patients with violence (V.SC) from those without violence (NV.SC) [71]. The GC-TOF-MS was used, and metabolites’ identification was completed by retention index and mass spectrometry data. Nineteen differential plasma metabolites between the V.SC and NV.SC groups were revealed by Wilcoxon–Mann–Whitney rank sum test (Table 7). The metabolites set, including the ratio of L-asparagine to L-aspartic acid, vanillylmandelic acid and glutaric acid, achieved a good classifier for discriminating schizophrenia patients with violence (AUC = 0.808) [71]. 

## 4. Concluding Remarks

As mentioned above, schizophrenia is a complex psychiatric disorder that can be triggered by the dysregulation of various biochemical pathways and systems. Additionally, this review demonstrates that GC-MS is a powerful instrumental platform for investigating these pathways and systems dysregulation with the final goal of discovering specific metabolic biomarkers of the disorder for its early diagnosis. The specific metabolic biomarkers can also reveal the biochemical background of the disease and hint at the best effective treatment. However, despite the potential and importance of the problem, the number of relevant studies is limited, and our systematic search found only 34 publications published to date. Most of these were published before 2015 and have not been followed up. With the recent advances in instrumental analysis, untargeted metabolomics, and a recognized need for an objective, unbiased clinical test for schizophrenia diagnosis, this research can be and should be intensified. 

It is also important that there is an inconsistency between the biomarkers or biomarkers patterns revealed by different groups even in the same matrix (see Table 7). This may be due to differences in metabolite extraction procedures and/or inaccurate identification of metabolites, e.g., identification only by comparison of experimental and library mass spectra can be misleading and should be augmented by the comparison of experimental and calculated/library compound retention index or using pure standards. The latter is more expensive, and sometimes pure standards are not available. In order to overcome the observed inconsistency, cooperation for standardization of the applied procedures and improvement of the experiment’s methodological quality should be developed. 

From our review, it can also be concluded that there is no systematic approach to finding accurate, unbiased markers of schizophrenia and that there is ample opportunity and need to intensify this research. For example, there is practically absent research on volatile organic compounds in the exhaled breath of schizophrenia patients. Breath research is now at the forefront of the search for specific volatile markers for the diagnosis of a plethora of diseases [72]. Breath analysis is very attractive for population screening as breath sampling is completely non-invasive, readily available and does not require patient transfer to hospitals and other health facilities, but it still has some limitations mentioned above (Section 3.1) [12].

Moreover, there are only a few studies on the holistic metabolite profiling in other biological matrices of schizophrenic patients, e.g., serum or sweat. Metabolite profiling is thought to be a useful tool for diagnosing disease and monitoring the effectiveness of its treatment. It is hoped that the information from metabolite profiling will make it possible to suggest personalized therapies for more effective treatment of the disease.

Thus, our review can clarify future directions in schizophrenia biomarkers research. Due to the importance and complexity of the early schizophrenia diagnosis, the search for unbiased specific biomarkers should be intensified. The indicated in this review inconsistencies presented by different teams’ data should be resolved by coordinated research of different teams and the development of harmonized procedures of the analysis, e.g., sample treatment, instrumental analysis, data treatment and interpretation. At present, the wide spectrum of the applied sampling and sample treatment method obviously leads to discrepancies in the detected biomarkers. More attention should be given to the biomarkers distinguishing schizophrenia from other mental diseases. 

## Figures and Tables

**Figure 1 molecules-28-00324-f001:**
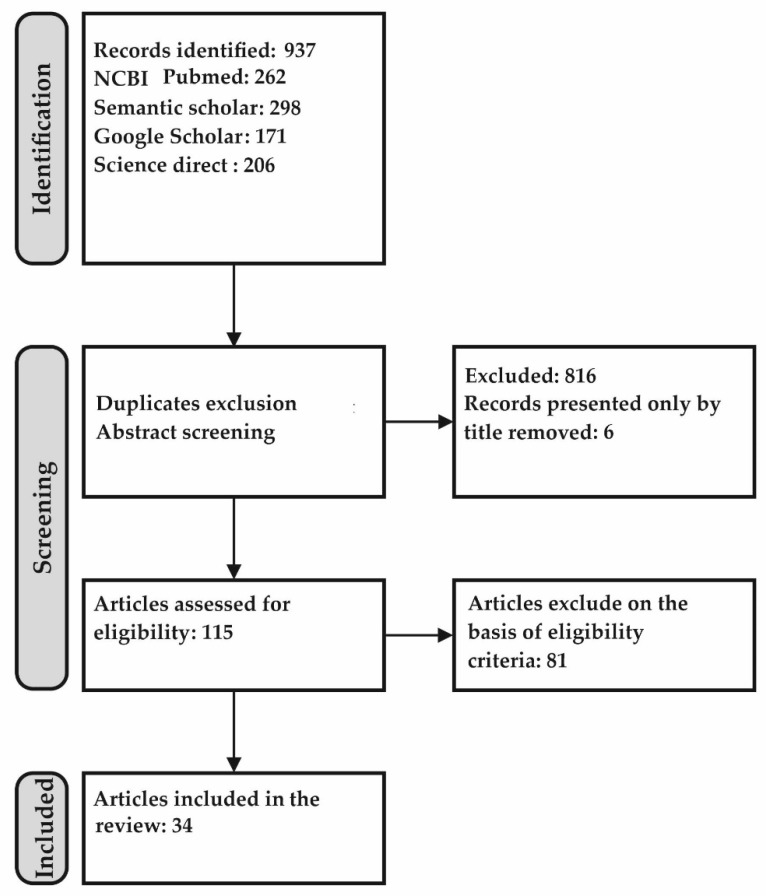
Flow chart of publication selection for the review.

**Table 1 molecules-28-00324-t001:** Studies of excreted volatile organic compounds for the identification schizophrenia markers.

Reference	Population Sample Schizophrenic/Other Mental Disease/Healthy	Biosample	VOCs	Claimed Outcomes
[15]	14/0/0	Sweat	TMHA	Detected and identified in schizophrenics
[17]	11/0/0	Sweat	TMHA	Failure to detect in schizophrenics
[18]	7/0/12	Sweat	TMHA	Detected in schizophrenics and controls. Cannot be considered as a marker of disease
[23]	25/26/37	Exhaled air	Pentane, Carbon disulfide	CS_2_ concentrations discriminate schizophrenics from other mental patients and healthy controls, pentane only from healthy controls
[26]	25/26/37	Exhaled air	2-Methylbutane Trichlorofluoromethane 2-Pentanol Pentane Dichloromethane Trichloroethene Benzene 1-Chloro-2-methylbutane 2,3,3-Trimethylpentane 2,2-Dimethylbutane Tetrachloroethene	Patients with schizophrenia can be distinguished from two other groups with a sensitivity of 80.0% and a specificity of 61.9%.

**Table 2 molecules-28-00324-t002:** Differentiating amino acids and amino acid metabolites between schizophrenic patients and healthy controls.

Reduced Compared to Control (Multiplicity, FC *)/Matrix	Reference	Increased Compared to Control (Multiplicity, FC)/Matrix	Reference
2-Aminoadipic acid (−1.27), urine	[33]	2-Aminobutyrate (1.28), serum	[33]
Aspartate (−2.20), serum	[32]	2-Aminobutyric acid (1,45), urine	[33]
Catechol (−1.83), urine	[33]	2-Hydroxybutyrate (2.45), serum	[33]
Cystine (−1,36), serum	[33]	5-Oxoproline (1.35), serum	[33]
Glycine (−1,8), serum	[32]	Aspartate (1.38) serum	[33]
Glycocyamine (−1.89), urine	[33]	Cystine (1.54), urine	[33]
N-Acetylaspartate (−1.96), serum	[32]	Glutamate (1.35), urine	[33]
Tryptophan (−2.07), serum	[32]	Glutamate (1.63), serum	[33]
		Isoleucine (1.3), urine	[33]
		Phenylalanine (1.14), serum	[33]
		Pipecolinic acid (1.65), urine	[33]
		Pyroglutamic acid (1.25), urine	[33]
		Serine (1.13)	[33]
		Tyramine, (>2) hair samples	[36]
		Valine (1.07), urine, plasma	[33,37]

* FC—fold change, a positive value of fold change indicates a relatively higher concentration of metabolites while a negative value of fold change indicates a relatively lower concentration in schizophrenic patients as compared to healthy controls.

**Table 3 molecules-28-00324-t003:** Differentiating between schizophrenic patients and healthy controls carbohydrate and TCA cycle metabolites.

Reduced Compared to Control (Multiplicity, FC)/Matrix	Reference	Increased Compared to Control (Multiplicity, FC)/Matrix	Reference
Citrate (−2.66), serum	[32,38]	Citrate (1.45), serum	[33]
Citrate (−0.71), PBMCs	[38]	cis-Aconitic acid (1.28), urine	[33]
α-Ketoglutarate (−1.52), serum	[32]	2-Oxoglutarate (1.59), serum	[33]
Lactic acid, PBMCs	[38]	Lactate (1.24–2.32), serum	[32,33]
Octanoic acid (−0.85), plasma	[37]	Malate (1.57), serum	[33]
Dihydroxyacetone phosphate (−1.28), PBMCs	[38]	Creatinine (0.35), plasma	[37]
Glycerol 3-phosphate (−0,49), PBMCs	[38]	Pyruvate (0.82), PBMCs	[38]
Glyceraldehyde-3-phosphate (−1.05), PBMCs	[38]	Pyruvate (1.88), serum	[33]
1,3-Bisphosphoglycerate (−1.61), serum	[32]	fructose (0.48), PBMCs	[38]
		fructose 6-phosphate (1.06), PBMCs	[38]
		Fumaric acid (0.24), plasma	[37]
		Maltose (0.51), plasma	[37]
		Glucose (0.67–1.50), serum, PBMCs	[32,38]
		Glucose 6-phosphate (1.48), PBMCs	[38]
		Sorbitol (0.31), plasma	[37]
		Succinic acid (0.76), PBMCs	[38]

**Table 4 molecules-28-00324-t004:** Differentiating between schizophrenic patients and healthy controls lipid and fatty acids metabolites.

Reduced Compared to Control (Multiplicity, FC)/Matrix	Reference	Increased Compared to Control (Multiplicity, FC)/Matrix	Reference
2,3-Dihydroxybutanoic acid (−1.3), urine	[33]	3-Hydroxybutyric acid (1.37), urine	[33]
Arachidonic acid, RBC	[40,41,43]	2-Hydroxybutyric acid (1.41), urine	[33]
Cholest-3,5-diene, serum	[34]	Tetradecanoic acid (1.45), serum	[33]
Cholest-5-en-3-ol, serum	[34]	4-Pentenoic acid (1.54), urine	[33]
Docosahexaenoic acid, RBC	[40,41,43]	3-Hydroxysebacic (5.55) acid, urine	[33]
Ethoxy- cholest-5-ene, serum	[34]	Glycerate (2.57), serum	[33]
Heptadecanoic acid, serum	[34]	Suberic acid (1.59), urine	[33]
Hydroxyacetic acid (−1.36), urine	[33]	β-Hydroxybutyrate (2.61), serum	[33]
Palmitic acid (−1.77), serum	[32]	Palmitic acid, RBC	[41]
Pentadecanoic acid, serum	[34]	Threonic acid (1.21), urine	[33]
Stearic acid (−1.81), serum	[32]	2-Ethyl-3-hydroxypropionic acid (1.29), urine	[33]
Eicosanoic acid, serum	[34]	Eicosenoic acid (1.96), serum	[33]
Linoleic acid (−2.69), serum	[32]	Linoleate (1.18), serum	[33]
Oleic acid (−2.52), serum	[32,34]	Oleic acid (2.09), serum	[33]
Glycerol (−0.35), plasma	[37]	Glycerol (1.42), serum	[32]
Cholesterol	[34]	Cholesterol (1.43), serum	[32]
		Hexadecanoic acid (1.4), serum	[33]
		3-Hydroxyadipic acid (2.06), urine	[33]
		Octadecanoic acid (1.14), serum	[33]

**Table 5 molecules-28-00324-t005:** Differentiating between schizophrenic patients and healthy controls neurotransmitters and their metabolites.

Reduced Compared to Control (Multiplicity, FC)/Matrix	Reference	Increased Compared to Control (Multiplicity, FC)/Matrix	Reference
2-Piperidinec carboxylic acid, serum	[34]	5-HO-DMT	[56,57,58]
6-Deoxy-mannofuranose > (−60), serum	[34]	Dopamine (0.31), plasma	[37]
Aspartic acid (−1.05), plasma	[37]	N,N-dimethyltryptamine	[59]
Homoserine (−0.50), plasma	[37]		
Oxoproline, serum	[34]		

**Table 6 molecules-28-00324-t006:** Oxidative stress biomarkers of schizophrenia.

Reduced Compared to Control (Multiplicity, FC)/Matrix	Reference	Increased Compared to Control (Multiplicity, FC)/Matrix	Reference
Hydroxylamine (−0.29), plasma	[37]	Pyroglutamic acid (0.13), plasma	[37]
γ-Tocopherol (−1.53), serum	[32]	γ-Tocopherol (0.34), plasma	[37]
		α-Tocopherol (0.19), plasma	[37]
		Serum SOD	[64]
		Urine 8-iso-PGF2α/creatinine (T0)	[64]

**Table 7 molecules-28-00324-t007:** Distinct metabolites between unmedicated schizophrenia patients and normal controls revealed by different metabolite profiling studies *.

Study	[32]	[33]	[37]	[34]
GC-MS platform	GC-Q-MS	GC-TOF-MS	GC-Q-MS	GC-Q-MS
Matrix	Serum	Serum	PBMCs	Serum
Discriminating metabolites	Glucose 1,3-Bisphosphoglycerate Lactate **Citrate** α-Ketoglutarate **Allantoin** Uric acid γ-Tocopherol N-Acetylaspartate Aspartate Glycine Tryptophan **Myo-inositol** Glucuronic acid Linoleic acid Oleic acid Stearic acid **Palmitic acid** Glycerol Cholesterol Lactobionic acid Erythrose	**Glycerate** **Eicosenoic acid** **β-Hydroxybutyrate** **Pyruvate** **Cystine** Malate Elaidic acid 2-Hydroxybutyrate Tetradecanoic acid Hexadecanoic acid Aspartate α-Oxo-pentanedioic acid Pyroglutamic acid Glutamate Citrate Phenylalanine Lactate Octadecanoic acid 2-Aminobutyrate Cholesterol Linoleic acid myo-lnositol	Octanoic acid Fumaric acid Valine Creatinine Inositol **Sorbitol** Maltose Hydroxylamine **Pyroglutamic acid** Tocopherol-g **Tocopherol-α** Aspartic acid Homoserine Dopamine Benzoic acid 2-Hydroxyethyl palmitate Glycerol Methyl Phosphate	1-Oxo-proline 2-Piperidinec carboxylic acid 6-Deoxy-mannofuranose Galactose oxime Oleic acid Pentadecanoic acid Heptadecanoic acid Eicosanoic acid Cholesterol
AUC, training set	-	**0.945**	0.82	-
AUC, test set	0.958	**0.895**	0.71	0.76–0.93 *

In bold, the biomarkers set with indicated below classification efficiency. *—AUC is defined for each metabolite separately.

## Data Availability

The data presented in this study are available on request from the corresponding author.

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
