# Peer review of "Experiences and Perspectives of GC-MS Application for the Search of Low Molecular Weight Discriminants of Schizophrenia"

_molecules, 2022, doi:10.3390/molecules28010324_

Round 1

Reviewer 1 Report

Authors summarize the studies on molecular biomarkers of schizophrenia utilizing gas chromatography – mass spectrometry as a versatile method offering high sensitivity and accuracy in untargeted analysis (imho: indeed but only if used properly). They refer to the most important results and try to outline future perspectives. The later needs to be improved.

Bellow please find the list of issues which needs corrections:

           1) Introduction:

a)       when writing that analysis of volatile metabolites is investigated nowadays for the development of non-invasive methods of preliminary screening and early diagnosis, authors should add at least a short information about the drawbacks of this approach (factors affecting analysis of human VOCs).

2) Methods:

a)      the criteria for inclusion the research articles in this review seem to be correct but some of them need further clarification – e.g. in #4 authors should explicitly define the “endogenous origin”, as in the present form they exclude only pharmaceuticals and narcotic substances. What about substances known as air-pollutants, tobacco-smoking or food-derived, etc.?

b)      in some cases authors are not consistent with their own inclusion criteria, as halons such as trichlorofluoromethane and other polychlorinated hydrocarbons are certainly not of endogenous origin, yet authors discuss these VOCs (ref. #26) in this review.

c)      The identification of target compounds must be amongst the inclusion criteria, publications speculating on “unidentified signals” shall not be discussed here.

3) Results:

The role of sample collection and preparation techniques should be emphasized stronger. It is a well-known factor seriously affecting the results of a GC-MS analysis. It was demonstrated in numerous original research papers and reviews that in the case of breath gas analysis (but also e.g. saliva or urine samples) it is of utmost importance to use the correct sample collection and preparation technique according to the optimized protocol. Otherwise, chemical composition of the sample changes and selection of “biomarker” based on quantitative comparison of its abundance in respective groups of subjects is completely spoiled. I suggest adding one chapter organized according to the most frequently used preparation protocols for sample matrices cited in this review (breath, serum, urine, etc.)

The identification of reported “biomarkers” has to be considered either in inclusion/exclusion criteria or at least needs to be clearly stated in each table mentioned in the paper (along with reference number), e.g. (a) MS Spectra .

-          I completely agree that identification of putative “markers” in numerous publications is either not provided or it is done not convincingly enough. The possibility of compound identification in GC-MS analysis is a great benefit over LC-MS or sensor arrays, but a naive sole library spectra match is often insufficient and without confirmation of chromatographic parameters for reference substance it may lead to a incorrect identification of an analyte, hence “discovery” of a false “biomarker”. Therefore, I do not understand why authors accept in this review publications in which volatile compounds were not identified at all (even preliminary, by spectra match) – these should be removed as not reliable (e.g. Ref. #27 by diNatale 2005).

-          Concerning the origin and the nature of discussed “biomarkers”: basically, authors refer to volatile compounds originating from diverse disorders, which should not be linked to schizophrenia by force. For instance, nearly all bacteria release carbon disulfide (CS2), including pathogens causing community-acquired pneumonia (Streptococcus pneumoniae, Haemophilus influenzae) or ventilator-associated pneumonia (Klebsiella pneumoniae, Pseudomonas aeruginosa and many, many other), all of which have absolutely no relation to schizophrenia. Similarly to hydrocarbons (including butane, pentane and their methylated derivatives), as the oxidative stress responsible for their production occurs in inflammation during numerous unrelated illnesses, including for instance heart failure. Hence, it needs to be clearly stated that so unspecific compounds simply cannot be used as biomarkers of schizophrenia.

-          On the basis of contradictory results concerning trans-3-methyl-2-hexenoic acid (TMHA) authors exemplify what a mess in scientific community can be done when speculations based on incorrect experiments are done and false biomarkers are announced. This is very good. Nevertheless, authors should take a stand in this dispute and express their opinion. Is there any reason (e.g. in methodology) underlying this discrepancy? Are there publications of other authors about TMHA analysis in schizophrenic patients that could help to solve this dispute?

-          Authors write that Yang et al. [32] used the same sample preparation procedure as Xuan et al. [31], however, this procedure is not explained here…

-          Similarly to the case of TMHA, also in this chapter authors present the contradictory results (here: glycine) – are there further publications of other authors about glycine analysis in schizophrenic patients?

-          There is an error in numbering the chapters – 3.2.4. is mentioned twice

4) Concluding remarks:

Authors mentioned in several sections that “the number of relevant studies is limited”. What is the reason for such a justification? The clear and detailed recommendations for future studies should be provided. Hereby, I would be very careful with the breath analysis, as this field of research is particularly capable for contradictory findings resulting from missing standardization at practically each analytical step, concomitant factors (yielding so called voodoo correlations) and inappropriate study design.

Reviewer 2 Report

The review is well prepared. I like the arrangement of the chapters according to the types of analytes. I appreciate an appropriate discussion in the event that the individual results do not correspond. I recommend publishing the article after a minor revision.

In the introduction, the criteria for inclusion of publications in the review are defined. The applied rules are also shown in Figure 1. Fig. 1 needs a correction. The given numbers do not correspond to each other. 262 + 298 + 171 + 206 = 937 (not 924); 115 - 84 = 31 (note 34) The only thing I understand: 924 - 803 - 6 = 115. 

Section 3.2.3 does not mention whether free fatty acids or bound fatty acids were analyzed. (Derivatization is required in both cases, but the results may differ significantly). For a better understanding, I recommend writing this information to citated publications (Only with [45] was mentioned).

Formal errors: * in Fig.1 ...not explained in the legend below the figure. The list of abbreviations is not complete (for example, AH, NH/AS, NP, EOS, Q, TOF missing). The abbreviation CSF is introduced on page 2 without explanation. The abbreviation is also in the list of abbreviations, but  the full name "cerebrospinal fluid" is used in the text. Review all abbreviations, list of abbreviations and their usage.

... by Xuan et al. ([31], Table XXX). This inconsistency demonstrates.....Table XXX.....?
